# Study on Size Optimization of a Protective Coal Pillar under a Double-Key Stratum Structure

**Daming Zhang, Hui Zhao * and Gang Li**

College of Mining Engineering, Liaoning Technical University, Fuxin 123000, China
* Correspondence: zhao923hui@163.com; Tel.: +131-3200-2057

**Abstract:** Because of the problem that the size of a coal pillar is difficult to determine and it is easy to waste coal resources, taking the Sangou Xindu coal industry as the engineering background, this paper examined the roof cutting and pressure relief technology of small coal pillar gob-side entry driving by means of theoretical analysis, laboratory experiments, and numerical simulation. Through theoretical calculations, the coal pillar width should not be less than 7.15 m without cutting the top. According to the theory of key strata, there are two key strata in the overlying strata on the working face, namely the main key strata and the subkey strata. Through numerical simulation, the stress evolution characteristics of the coal at the side of the goaf under the double-key stratum roof cutting structure are studied. When the roof is not cut, the roof cutting height is 6 m and 12 m and the width of the lateral stress reduction zone in the goaf is 4 m, 8 m, and 10.5 m, respectively. Therefore, the cutting top height is determined to be 12 m and the hydraulic fracturing cutting plan is designed. After cutting the top of the main key stratum, the size of the coal pillar is optimized. Three schemes of coal pillar sizes of 6 m, 5 m, and 4 m are designed for simulation and the analysis shows that it is most reasonable to keep a 5 m wide coal pillar. After field application, the deformation of the surrounding rock in the roadway is within a controllable range and the roadway use is good.

**Keywords:** double-key stratum; cutting roof and pressure relief; small coal pillar; numerical simulation; driving roadways along goaf

## 1. Introduction

In recent years, recoverable coal resources have been decreasing daily. For the coal industry, reducing the loss rate of coal mining while ensuring safety, high production, and high efficiency is an urgent problem to be solved in green coal mining [1–4].

In a typical key stratum structure, the mining stress distribution law of the underlying strata is mainly related to the location and thickness of the key stratum [5]. Li Guodong [6] concluded that there are two key layers, upper and lower, above the roof of working face according to the in situ rock pressure observation and the criterion of the key layer. After the two key layers are broken, a "double masonry beam" structure is formed, and its pressure has a significant periodicity phenomenon. In underground mining, applying gob-side entry driving technology can improve the coal recovery rate [7,8], and a reasonable coal pillar width is the key to ensuring the success of gob-side entry driving. The determination of the reasonable width of coal pillar in gob-side entry is affected by many factors. Li Xuehua et al. [9] summarized and analyzed that the strength and thickness of the coal seam, support strength, and width of the narrow coal pillar are the key factors affecting the stability of a narrow coal pillar. Zhao Min, Wang Yi, and Liu Hongyang et al. [10–12] determined the reasonable width of a narrow coal pillar in gob-side entry driving by theoretical analysis and numerical simulation. However, there has not been any complete research method that can take all of the influencing factors into account to determine coal pillar width. Compared with the traditional large coal pillar, the optimized small coal pillar greatly saves coal resources to ensure safe production [13,14].

In the third revolution of mining science and technology, the Academician He Man-chao et al. proposed the theory of "cutting roof short wall beam", based on the cutting roof and pressure relief automatic roadway formation without coal pillar mining technology [15]. Yang Xiaojie et al. [16] proposed the method of "adjacent roadway roof cutting and pressure relief", which provided a new idea for controlling roadway deformation, transforming from "strong support" to "pressure relief". At present, roof cutting and pressure relief technologies include prefracturing and blasting technology, hydraulic fracturing technology, $CO_2$ gas fracturing technology, and so on [17–19]. In contrast, hydraulic fracturing technology has lower costs and higher safety [20].

Many scholars [21–24] have concluded that the cutting parameters are the keys to achieving accurate cutting pressure relief. At present, when designing the top cutting height, most of the top cutting position is controlled within the range of the immediate roof, and the roof pressure relief is not sufficient. How to accurately control the cutting height to achieve fully optimize the stress of goaf roadway needs further discussion. In addition, the lateral roof cutting of goaf roadway changes the strata breaking form, which makes the movement characteristics and breaking laws of overlying rock structure different from the conventional conditions. However, research on this condition is relatively lacking.

## 2. Methods

### 2.1. Engineering Background

Jinsheng Sangou Xindu Coal Industry of the Shanxi Jin Coal Group is located in the southwest of Qinshui County, Jincheng City, Shanxi Province, and the administrative division is under the jurisdiction of Longgang Town, Qinshui County. Its geographical coordinates are 112°01′32″~112°04′05″ east longitude; north latitude 35°38′43″~35°39′45″. The location map is shown in Figure 1. The #2 coal seam of the Shanxi Group and the #15 coal seam of the Taiyuan Group in this mining area are recoverable coal seams. The good field is generally an anticline structure with a gentle dip angle between 4° and 14°. The thickness of #15 coal is 0.75 m~3.80 m, with an average of 2.45 m. It is a medium-thick coal seam. The 1301 fully mechanized mining face is adjacent to fault F6 in the east and north for solid coal and south of the 1203 lane. In the west of the mining area, the track roadway has a strike length of 830 m and an average inclined length of 156 m. The layout of the working face is shown in Figure 2. After the mining of working face 1301, working face 1201 is mined (its location is not listed owing to drawing restrictions), and the mining roadway of working face 1203 is driven while mining working face 1201. The original coal pillar of the other working faces is set at 25 m. In the mining process, the 1301 working face has been subjected to the roof cutting operation on the 1301 ventilation roadway. This project has researched the gob-side entry technology of the small coal pillar in the 1203 working face. The bar chart of the working face is shown in Figure 3.

### 2.2. Build a Model

FLAC$^{3D}$5.0 software was used for numerical simulation, taking the Sangou Xindu coal industry as the engineering background, to study the reasonable size of small coal pillars in the 1203 working face. The length of the model is 430 m, the height is 100 m, the length of the 1301 face is 160 m, and the mining height is 2.6 m. The constraint displacements at the model's left and right and lower boundaries were zero, the top of the model is a free boundary, and an equivalent load of 5.25 MPa was applied to the top. In order to improve the computational efficiency, the local mesh density of the study object is 0.25 m/grid and the rest of the mesh density is 1 m/grid. The model diagram is shown in Figure 4. Goaf parameters are shown in Table 1.

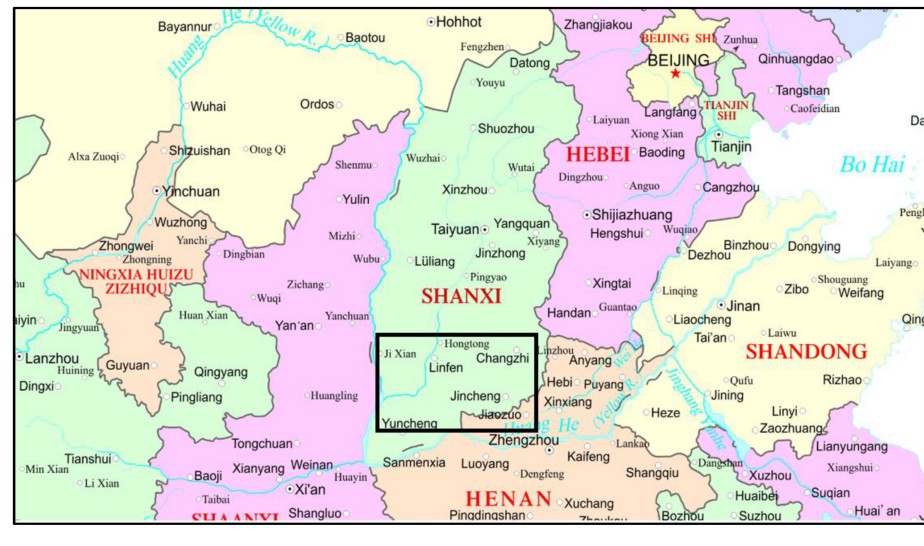

(**a**)

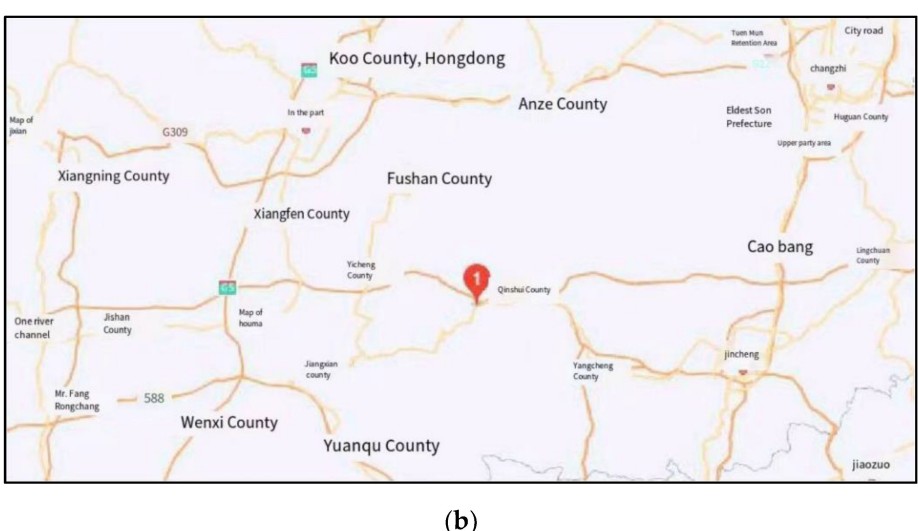

(**b**)

**Figure 1.** Geographical location diagram of the Sandou Xindu coal industry: (**a**) partial map of China; (**b**) locally enlarged map.

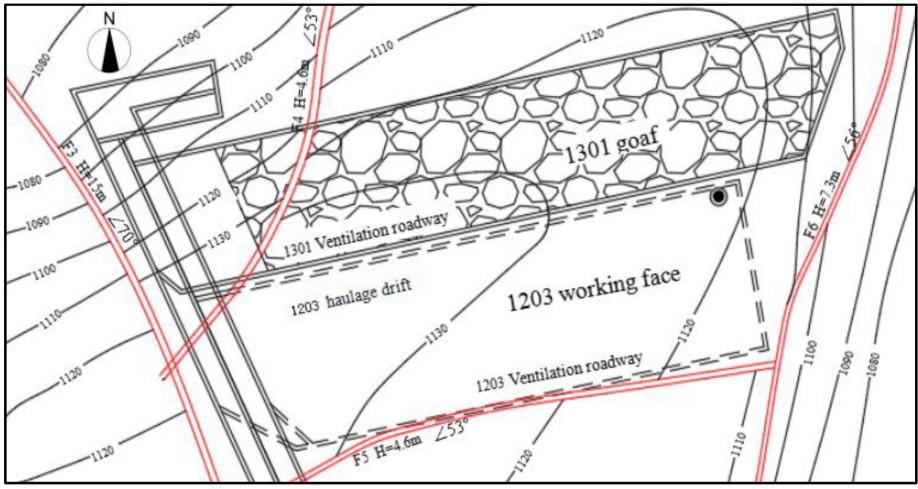

**Figure 2.** Working face layout drawing.

| Name of rock | thickness | | columnar | | Description of lithology |
|---|---|---|---|---|---|
| K3 limestone | 1.3 | | | | Dark grey, local facies to sandy mudstone |
| mudstone | 0.5 | | | | Gray-black, thin layer, dense, soft |
| K2 limestone | 1.3 | | | | Dark grey with lots of flint bands and nodules |
| mudstone | 2.8 | | | | Gray-black, medium-thick laminate, dense, soft |
| K2 limestone | 5.3 | | | | Dark gray, thickly layered, Containing a large number of flint bands and nodules |
| mudstone | 0.3 | | | | Gray-black, thin layer dense, soft |
| 15# coal | 2.6 | | | | Black, locally separated gangue |
| mudstone | 0.9 | | | | Gray black, dense, soft |
| bauxitic mudstone | 3.7 | | | | Light gray, medium - thick layer, slightly silt |

**Figure 3.** Comprehensive bar chart of the 1301 working face.

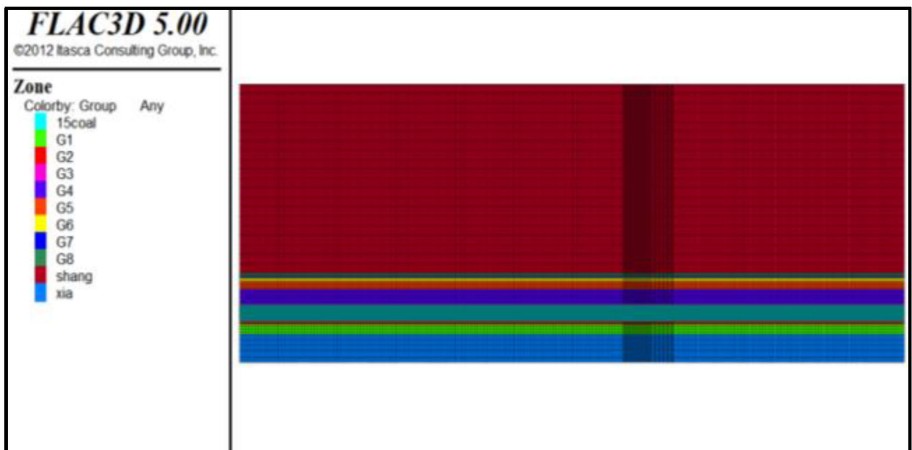

**Figure 4.** Model schematic diagram.

**Table 1.** Material parameters of goaf.

| Density/[kg·m$^{-3}$] | Bulk Modulus [GPa] | Shear Modulus [GPa] | Dilatancy Angle [°] | Frictional Angle [°] |
|---|---|---|---|---|
| 1700 | 10 | 6 | 10.3 | 30 |

*2.3. Theoretical Calculation of the Sectional Coal Pillar Retaining Size under Noncut Roof Conditions*

The reasonable width of the coal pillar is a critical factor for roadway stability. The theoretical model of a reasonable width of a coal pillar in roadway protection is shown in Figure 5, and the calculation formula is

$$B = x_0 + x_1 + x_2 \tag{1}$$

where $x_0$ is the width of the plastic zone of the coal pillar; $x_1$ is the length of the bolt (cable); and $x_2$ is the coal pillar safety zone, generally 0.15~0.35 ($x_0 + x_1$).

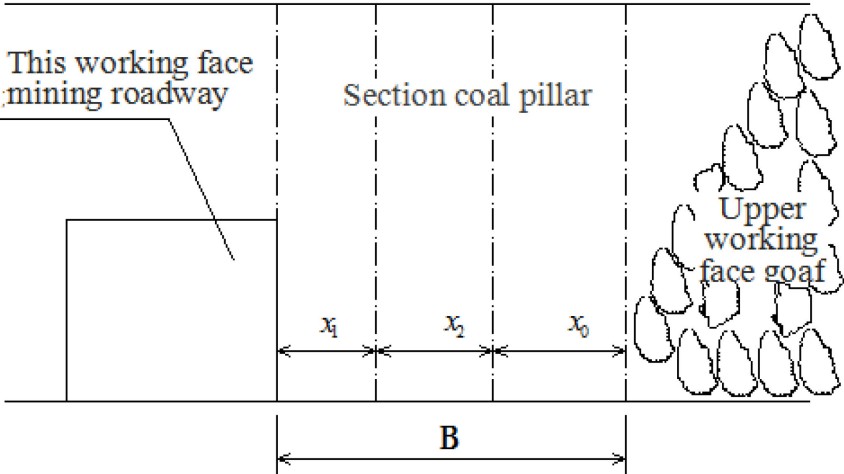

**Figure 5.** Calculation diagram of the small coal pillar size in the section.

According to the geological report of the #15 coal seam, the coal seam is buried at a depth $H = 300$ m; the average bulk density of the overlying strata $\gamma = 26$ kN/m$^3$; the pillar height is the height of the mining roadway $m = 2.8$ m; and the cohesion and internal friction angle at the junction of the coal pillar and roof and floor are $C_0 = 0.82$ MPa and $\phi = 25°$, respectively. The supporting resistance of the coal pillar on one side of the goaf is $P_X = 0$; the coefficient of the stress concentration $K = 2.3$; the Poisson's ratio for coal #15 is $\mu = 0.31$; the lateral pressure coefficient $\lambda = \mu/(1 - \mu) = 0.45$; and the length of the bolt of the coal pillar at the side of the roadway $x_1$ is 2.0 m. The above data are substituted into the following equation:

$$x_0 = \frac{m\lambda}{2\tan\phi_0} \ln \frac{K\gamma H + \frac{C_0}{\tan\phi_0}}{\frac{C_0}{\tan\phi_0} + \frac{p_x}{\lambda}} \tag{2}$$

Then, $B = x_0 + x_1 + x_2 = 3.3 + 2.0 + 0.35 \times (2.7 + 2.0) = 7.15$ m.

Through the above calculation, it is theoretically analyzed that the minimum size of the section coal pillar should not be less than 7.15 m when roof cutting measures are not taken to relieve pressure. If the roof cutting operation is taken, the size of the section coal pillar can be further reduced.

### 2.4. Key Strata Discrimination of the Working Face

In the overlying strata moving from the bottom to top, sometimes, two or more layers of strata move synchronously, called composite strata. According to the relevant theory of the compound effect of rock strata [25], G6, G7, and G8 can be determined as compound rock strata, which are named Ge. The physical and mechanical parameters of #15 coal rock are shown in Table 2, from which one can calculate the equivalent thickness, equivalent moment of inertia, equivalent elastic modulus, and bulk density of composite rock strata.

$$h_e = h_6 + h_7 + h_8 = 3.1 \text{ m}$$
$$I_e = \frac{(h_6 + h_7 + h_8)^3}{12} = 2.482 \text{ m}^3$$
$$E_e = \frac{E_6 I_6 + E_7 I_7 + E_8 I_8}{I_e} = 677,130.76 \text{ MPa}$$
$$r_e = \frac{r_6 h_6 + r_7 h_7 + r_8 h_8}{h_e} = 26.55 \text{ KN/m}^3$$

**Table 2.** Determination results of the physical and mechanical parameters of the #15 coal rock.

| Serial Number | Rock Name | Density [kg·m⁻³] | Tensile Strength [MPa] | Modulus of Elasticity [MPa] | Poisson Ratio | Angle of Internal Friction [°] | Cohesive Strength [MPa] |
|---|---|---|---|---|---|---|---|
| G8 | K3 limestone | 2712 | 4.16 | 28,194 | 0.20 | 39°44′ | 8.15 |
| G7 | Mudstone | 2684 | 2.05 | 17,048 | 0.23 | 35°18′ | 3.67 |
| G6 | K2 limestone | 2712 | 4.16 | 28,194 | 0.20 | 39°44′ | 8.15 |
| G5 | Mudstone | 2684 | 2.05 | 17,048 | 0.23 | 35°18′ | 3.67 |
| G4 | K2 limestone | 2712 | 4.16 | 28,194 | 0.20 | 39°44′ | 8.15 |
| G3 | Mud stone | 2684 | 2.05 | 17,048 | 0.23 | 35°18′ | 3.67 |
| #15 coal | #15 coal | 1608 | 1.18 | 6107 | 0.29 | 30°21′ | 1.91 |
| G2 | Mudstone | 2684 | 2.05 | 17,048 | 0.23 | 35°18′ | 3.67 |
| G1 | Bauxitic mudstone | 2360 | 2.17 | 14,326 | 0.20 | 30°47 | 2.75 |

Based on the key strata theory, the "main key layer" controls all the strata from the coal seam roof to the surface, while the "sub-key layer" only plays a major role in controlling the overlying local strata. The following two conditions are required to determine the position of the critical layer [26,27]:

(1) Rigid condition discrimination

Assuming that the first stratum is a hard stratum, which controls the synchronous deformation of stratum 1~$m$, then stratum $m + 1$ is the second hard stratum. The overburden load of the first hard rock layer is

$$q_1(x)|_m = \frac{E_1 h_1^3 \sum\limits_{i=1}^{m} h_i r_i}{\sum\limits_{i=1}^{m} E_i h_i^3} \qquad i = 1, 2, \cdots m \tag{3}$$

where $q_{1/m}$ is the load on the first layer when calculated to the m layer, KPa; $h_i$ is the thickness of the $i$th layer, m; $r_i$ is the bulk density of the $i$th layer, KN/m³; and $E_i$ is the elastic modulus of the $i$th layer, MPa.

When Equation (4) is satisfied, the $m + 1$ stratum is a hard stratum:

$$q_1(x)|_m \, q_1(x)|_{m+1} \tag{4}$$

According to the above calculation formula and combined with Table 1, the calculation is carried out layer by layer from the bottom to top, starting from the G3 mudstone of the direct roof of #15 coal.

$$q_3 = r_3 h_3 = 7.56 \text{ kPa}$$

$$q_{3/4} = \frac{E_3 h_3^3 (r_3 h_3 + r_4 h_4)}{E_3 h_3^3 + E_4 h_4^3} = 0.0097 \text{ kPa}$$

$q_{3/4} < q_3$, so G4 is a hard rock stratum and the load of the G3 rock stratum is 7.56 KPa. The load of the overlying strata of G4 is calculated as follows:

$$q_4 = 140.98 \text{ kPa} \, ; q_{4/5} = 200.84 \text{ kPa} \, ; q_{4/e} = 50.35 \text{ kPa}$$

Then, the composite rock Ge is a hard rock and the load of G4 is 200.84 KPa.

(2) Strength condition discrimination

Only a few hard rock strata are found in the stiffness discrimination. The main key strata can be determined by comparing the fracture sequence of hard rock strata through

strength discrimination. The fracture distance of hard rock strata can be calculated by Equation (5):

$$L_i = h_i \sqrt{\frac{2\sigma_{ti}}{q_i}} \tag{5}$$

where $q_i$ is the load on hard rock, MPa, and $\sigma_{ti}$ is the tensile strength of hard rock, MPa. If layer $i$ +1 is the key layer, then

$$L_i < L_{i+1} \tag{6}$$

At this time, the $i$th layer is the subkey layer. If not, it is necessary to apply the $i + 1$ layer and all the rock layers controlled by it to the $i$th layer to perform the above calculation again.

Combined with the research results of Mao Liaoxing et al. [28], when the traditional ground pressure theory is used to estimate the ultimate broken length of key strata with composite key strata, the composite effect and the influence of the mining height should be considered. The following equation can be used to modify the ultimate broken length of composite strata:

$$L_e = kl_e - k_1(M - M_0) \tag{7}$$

where $L_e$ is the modified ultimate broken length of the compound key strata and $l_e$ is the ultimate broken length of the compound key strata calculated by the traditional method. $k_1$ is the increasing coefficient of the ultimate broken length of the compound key strata with the mining height with a value of 1.33. $M$ is the average mining height of the working face and $M_0$ is the height of the 2 m working face. The load of the composite strata is $q_e = \gamma_e h_e$ = 82.305 KPa and $\sigma_{te}$ is the tensile strength of limestone, which is 4.16 MPa. According to the above calculation formula, it can be concluded that

$$L_4 = 34.11 \text{ m } l_e = 31.17 \text{ m } L_e = 41.28 \text{ m}$$

According to the size of the broken length, it can be determined that the composite strata Ge 11.5 m above the coal seam is the composite main key strata and G4 5.6 m above the coal seam is the subkey strata.

## 3. Results and Discussion

*3.1. Numerical Simulation Analysis of Roof Cutting and Pressure Relief*

3.1.1. Analysis of the Surrounding Rock Stress Characteristics When the Roof Cutting Measure Is Not Taken

Figure 6 shows the cloud diagram of the surrounding rock stress distribution after 1301 working face mining, and Figure 7 shows the curve diagram of the lateral surrounding rock stress distribution after 1301 working face mining. Figures 5 and 6 show that the original rock stress of the no. 15 coal seam at the 1203 working face is approximately 7.5 MPa. After mining the 1301 working face, stress concentration is formed on both sides of the goaf, among which a stress reduction zone (4 m) and stress increase zone (82 m) are formed at the 1203 working face side. The peak value of the stress increase zone is 16.81 MPa, which is located 6 m away from the goaf boundary and the stress concentration coefficient is 2.24. If a small coal pillar and roadway are to be set at this place, the roadway will be in the stress increase area, which is not conducive to the stability of the roadway, so roof cutting and pressure relief measures should be taken at this place.

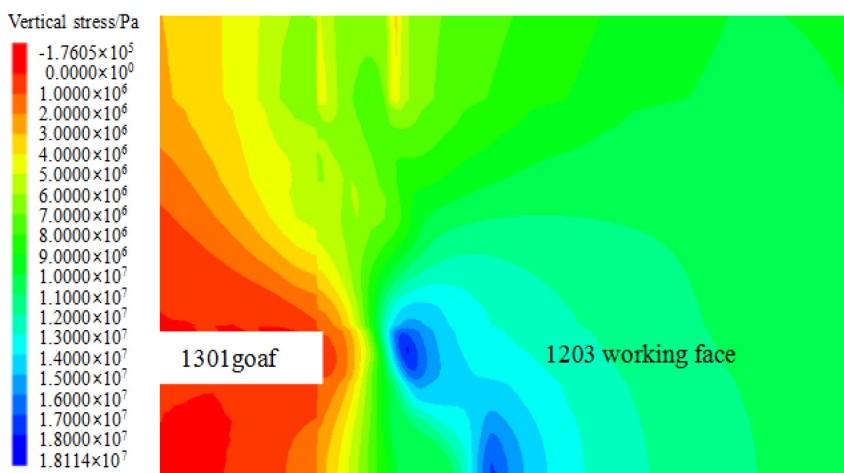

**Figure 6.** Vertical stress cloud diagram of the surrounding rock after mining the 1301 working face.

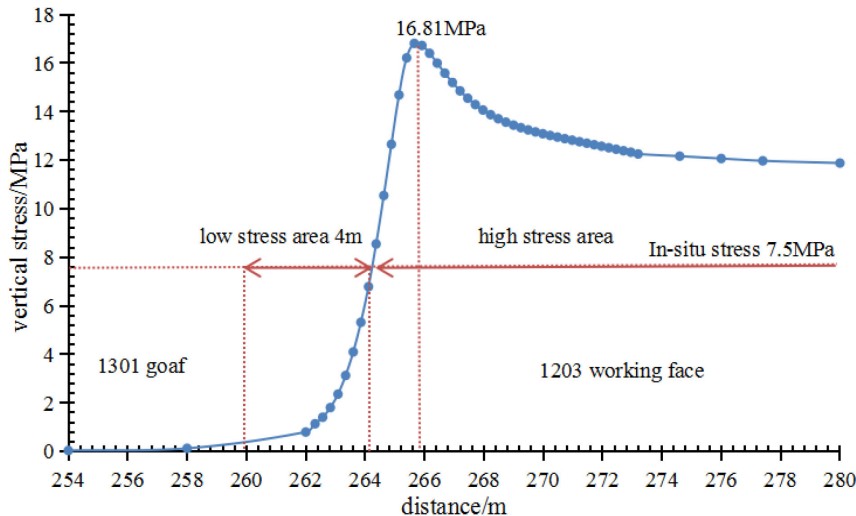

**Figure 7.** Local curve of the lateral surrounding rock vertical stress after mining the 1301 working face.

3.1.2. Analysis of the Stress Characteristics of the Surrounding Rock under Roof Cutting and Pressure Relief

The stress distribution characteristics of the surrounding rock are simulated under the conditions that the roof cutting height is 6 m (cut to the subkey strata) and the roof cutting height is 12 m (cut to the main key strata) to determine which key strata have more influence on rock mass activity. Figures 8 and 9 show that the range of the lateral stress reduction zone in the goaf significantly expands after roof cutting. When the cutting top is 6 m high, the peak stress in the coal pillar is 15.72 MPa and the range of the lateral stress reduction zone in the goaf is 8 m. If the coal pillar and roadway are arranged here, the width of the small coal pillar should be less than 3.8 m. The coal pillar size is small, it is easy to leak air to the adjacent goaf, and there is the risk of spontaneous coal combustion. When the cutting top is 12 m high, the peak stress in the coal pillar decreases to 14.49 MPa and the range of the lateral stress reduction zone in the goaf expands to 10.5 m. If the coal pillar and roadway are arranged here, the width of the small coal pillar should be less than 6.3 m. In conclusion, the pressure relief effect is better when the roof cutting height is 12 m.

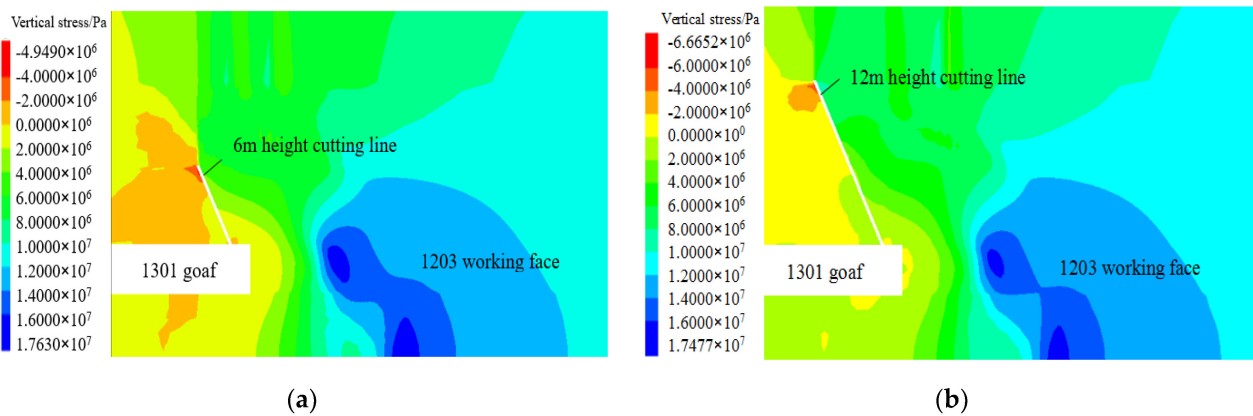

**Figure 8.** Vertical stress cloud map of the surrounding rock after roof cutting: (**a**) roof cutting height of 6 m (cut to the subkey strata); (**b**) roof cutting height of 12 m (cut to the main key strata).

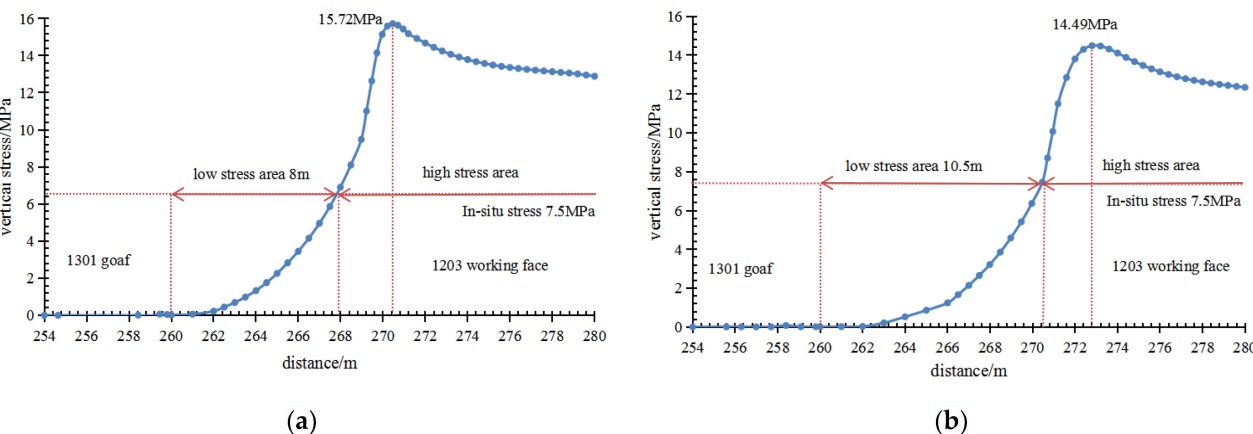

**Figure 9.** Diagram of the vertical stress distribution of the lateral surrounding rock of the 1301 goaf after roof cutting: (**a**) roof cutting height of 6 m (cut to the subkey strata); (**b**) roof cutting height of 12 m (cut to the main key strata).

The above numerical simulation analysis shows that the main key strata have a greater control effect on the mine pressure of the working face. Therefore, the roof cutting height is set to 12 m when the roof cutting operation is carried out.

### 3.2. Roof Cutting Scheme Design

In this paper, the hydraulic fracturing method is adopted for the roof cutting operation. According to the roadway layout in the working face, the hydraulic fracturing test hole is arranged axially in the 1301 ventilation roadway with a column pneumatic drill. The fracturing hole is arranged in a unilateral way, with an angle of 70° and a spacing of 10 m between the drill hole and the roadway, and it is set in the direction of the goaf. Fracturing and control drilling holes with diameters of $\varphi 46 \sim 48$ mm were constructed. An emulsion pump station is adopted, with a rated pressure of 31.5 MPa and a flow rate of 80 L/min. A cutting tool with $\varphi 38$ mm was used to cut the initial fracture. To ensure that the fracture could crack during cutting, the cutting depth should exceed 3 m, and the total height of the cutting roof should be 12 m.

### 3.3. Optimization of the Small Coal Pillar Size

As the lateral stress reduction zone of the goaf is 10.5 m when the roof cutting height top is 12 m, three schemes of coal pillars with different widths of 6 m, 5 m, and 4 m are selected for simulation. Figures 10 and 11 show that, the smaller the coal pillar size, the smaller the vertical stress in the coal pillar, and the peak stress is 4.58 MPa, 3.07 MPa, and

1.79 MPa, respectively. When the width of the coal pillar is 4 m, its maximum bearing capacity is 1.79 MPa, which is far less than the applied overburden load, which is not conducive to the long-term stability of the coal pillar. The surrounding rock is easily broken, so the size of the coal pillar should be greater than 4 m. For small coal pillars of different sizes, there is no noticeable difference in the stress distribution at the 1203 working face. The stress concentration zone is 3~10 m away from the right side of the roadway and the stress peak value is 16.4~16.62 MPa.

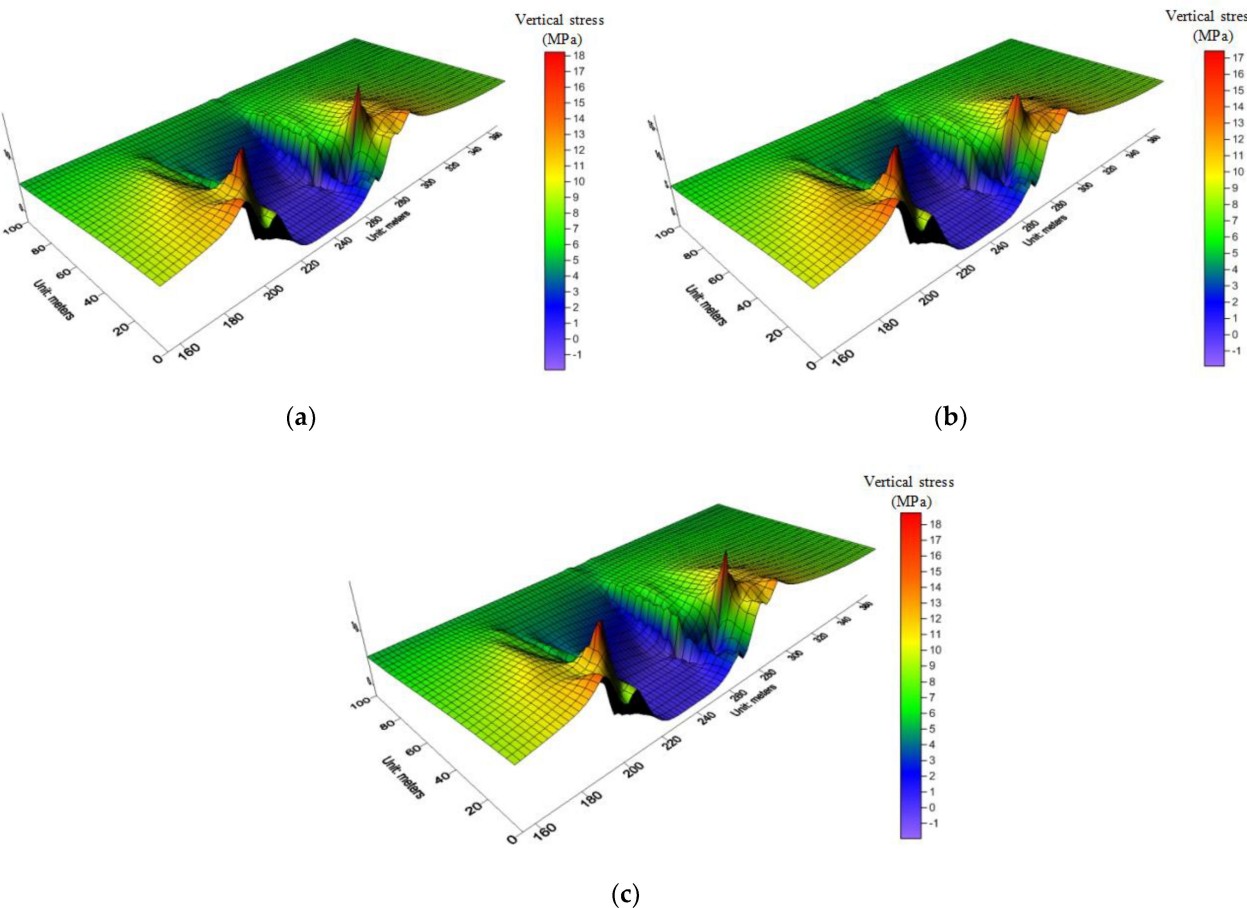

(**a**)　　　　　　　　　　　　　　　　　(**b**)

(**c**)

**Figure 10.** Three-dimensional (3D) curved surface of vertical stress of coal pillars of different sizes: (**a**) the width of the coal pillar is 4 m; (**b**) the width of the coal pillar is 5 m; (**c**) the width of the coal pillar is 6 m.

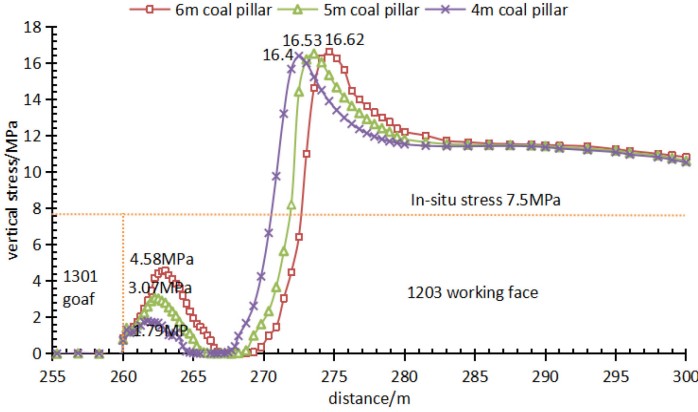

**Figure 11.** Diagram of the vertical stress distribution of the coal pillar of different widths.

The deformation curve of the roadway surrounding rock is shown in Figure 12. With the increase in coal pillar size, the moving amount of the two sides of the roadway increases and the moving amount of the roof and floor decreases. When the coal pillar size is 6 m, 5 m, and 4 m, the peak displacements of the two sides of the roadway are 309.98 mm, 255.79 mm, and 187.55 mm, respectively. The displacements of the roof and floor exhibit little change, and their peak values are 170.78 mm, 175.18 mm, and 171.51 mm, respectively. To prevent air leakage to the adjacent goaf, a 5 m coal pillar is more conducive to fire prevention than a 4 m coal pillar.

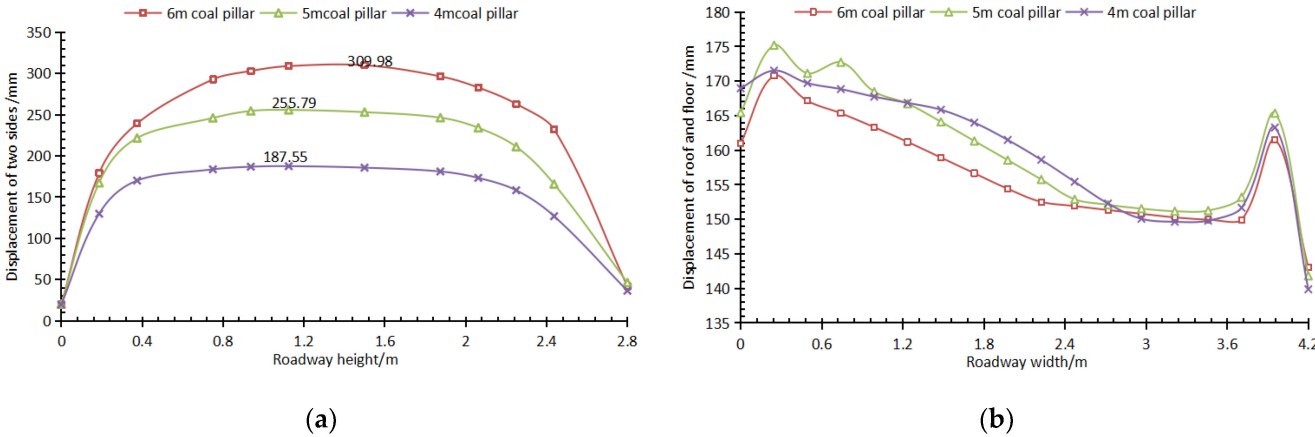

(**a**)                                                         (**b**)

**Figure 12.** Deformation curve of the roadway surrounding rock with coal pillars of different sizes: (**a**) displacement of two sides of the roadway; (**b**) displacement of the roof and floor of the roadway.

Considering the change law of stress in the coal pillar and the deformation characteristics of the surrounding rock comprehensively, it is reasonable to set a 5 m wide coal pillar under the premise of roof cutting and pressure relief treatment. This can ensure the stability of the roadway surrounding rock and save coal resources to the greatest extent.

## 4. Field Engineering Application

To monitor the coal pillar and the surrounding rock state of the roadway after roof cutting and pressure relief, two measuring stations are set in the roof cutting and pressure relief section. The first measuring station is located in the haulageway 100 m away from the open-off cut of the 1203 working face, and the second measuring station is 50 m away from the first measuring station.

In the field engineering application, to prevent air leakage from the coal pillar to the adjacent goaf, the coal pillar is sprayed with a thickness of 200 mm. According to the statistics of roadway deformation of the 1203 haulageway during tunneling, displacement monitoring data are obtained, as shown in Figure 13. With the continuous excavation of the gob-side entry, the deformation trend of the surrounding rock increases towards stability. The surrounding rock deformation of the roadway is mainly caused by right wall deformation and roof subsidence. The deformation of the right wall accounts for 60% of the deformation of the two sides and the roof deformation accounts for 87% of the displacement of the roof and floor. The deformation speed of the surrounding rock is fast, within 25 days of roadway formation. With time, the deformation speed of the surrounding rock slows down and the roadway enters a stable state. Currently, the maximum displacements of the roadway roof, floor, and left and right sides are 109.4 mm, 16 mm, 179 mm, and 122.8 mm, respectively. The above situation indicates that the roadway is in good use.

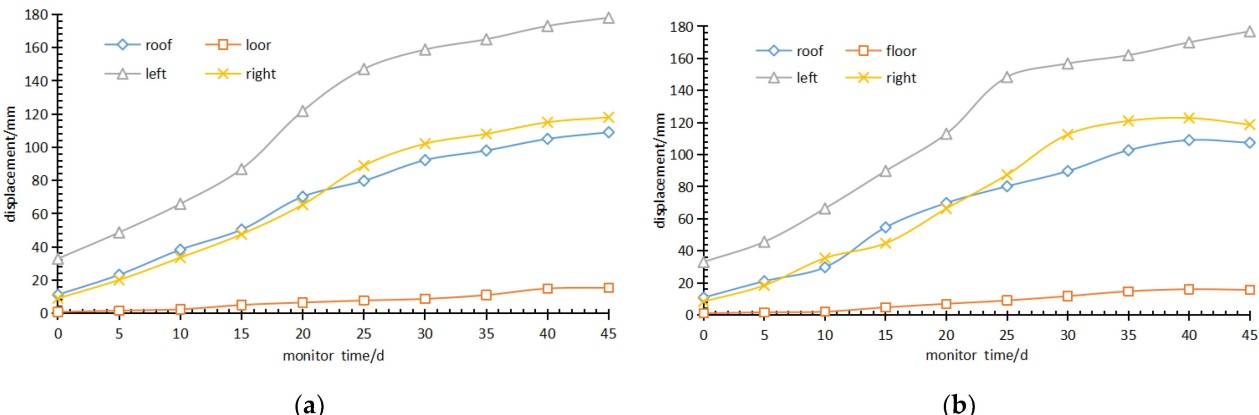

**Figure 13.** Deformation of the roadway surrounding rock during 1203 transport heading: (**a**) no. 1 station; (**b**) no. 2 station.

The vertical stress inside the 5 m small coal pillar is monitored on site and the overall stress distribution is symmetrical. The maximum stress in the center of the coal pillar is 4.42 MPa and the stress on both sides gradually decreases, as shown in Figure 14. It can be seen that the internal stress distribution of coal pillar after roof cutting and pressure relief is in good agreement with the results of the numerical simulation (see Figure 10b).

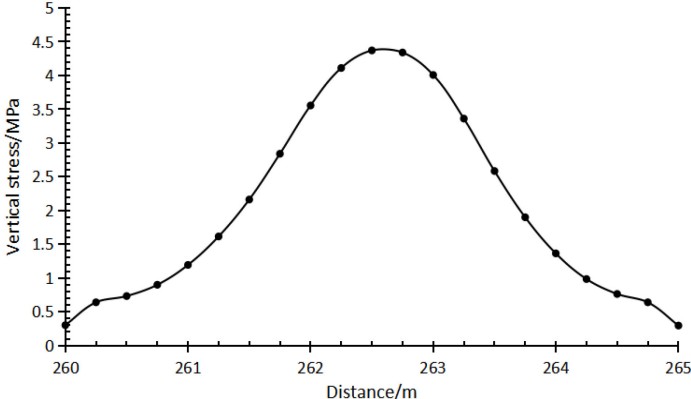

**Figure 14.** Stress monitoring curve of the small coal pillar.

## 5. Conclusions

- Through theoretical calculations, it is concluded that the width of the section pillar should not be less than 7.15 m without taking roof cutting measures. The size of the section coal pillar can be further reduced if the measure of roof cutting and pressure relief is taken. In addition, based on the theory of key strata, it is concluded that there are main and subkey strata in the overlying strata on the working face. Moreover, limestone G4, 5.6 m above the coal seam, is the subkey stratum, and the composite rock Ge, 11.5 m above the coal seam, is the composite main key stratum.
- The numerical simulation method is used to study the stress characteristics of the surrounding rock of the goaf side after the roof cutting of the main and subkey layers. It is concluded that the main key layer plays a main control role on the rock mass activities. When the top cutting height is 12 m, a stress reduction zone of 10.5 m is formed in the surrounding rock of the goaf side.
- Through the optimization study of small coal pillar size, it is concluded that the roadway surrounding rock is more conducive to stability when a 5 m coal pillar is set. After field application, the deformation of the roadway surrounding rock is monitored in a controllable range and the roadway is good in use.



**Author Contributions:** Conceptualization, D.Z. and H.Z.; methodology, D.Z. and H.Z.; software, H.Z.; formal analysis, H.Z.; investigation, H.Z.; data curation, H.Z.; writing—original draft preparation, H.Z.; writing—review and editing, D.Z.; visualization, H.Z.; funding acquisition, G.L. All authors have read and agreed to the published version of the manuscript.

**Funding:** This research was funded by the National Natural Science Foundation of China, grant number 51774165.

**Institutional Review Board Statement:** Not applicable.

**Informed Consent Statement:** Not applicable.

**Data Availability Statement:** The data presented in this study are available in the article.

**Conflicts of Interest:** The authors declare no conflict of interest.

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
