# Peer review of "Study on Size Optimization of a Protective Coal Pillar under a Double-Key Stratum Structure"

_applsci, doi:10.3390/app122211868_

Round 1
Reviewer 1 Report
This paper conducted a study on the cutting roof and pressure relief technology with a small coal pillar supporting roadway. A comprehensive research methods were used, including the analytical modelling, numerical simulation and experimental work. This study is quite interesting. I recommended the authors to consider the following issues and a revision is recommended.
(1) When the authors mentioned the green coal mining, the authors only used one reference to support this. It cannot fully support the corresponding topics, especially the ensuring safety which is mentioned by the authors. I think it will be better to add references regarding the coal mining safety to support this, such as 10.1016/j.conbuildmat.2022.127558, 10.1016/j.engfailanal.2022.106640.
(2) In the introduction, the authors mentioned the key stratum. But what is the key stratum. I recommended the authors to use a few sentences to give the basic introduction of the key stratum.
(3) In the manuscript, the expression of "CO2" is not accurate. In fact, the value of "2" should be subscript.
(4) When the authors give the reviewing of the previous references, I think a critical reviewing should be given. So, please use a few sentences to summarise and indicate the shortcomings in previous research.
(5) In Figure 2, in the column chart of the rock strata, the basic information and character of each rock strata should be given.
(6) In numerical simulation, when the authors used the equivalent load of 5.25 MPa, how is this value determined? I mean why should the authors use this value?
(7) Can authors explain how is the equation (2) is obtained? Or how is this equation deduced?
Reviewer 2 Report
1. Abstract must be able to show the problem statement, methodology, scope, and the results; keeping in view the journal instructions for authors.
2. Although the authors have discussed about main key strata and subkey strata in the abstract, they must discuss in details in the introduction section or in other appropriate section.
3. Include a map, showing the project location.
4. After discussing the literature, indicate the research gap.
5. Novelty of the paper.
6. Along with the engineering background, working schedule is also required.
7. Why the authors adopted the Mohr-coulm model in the numerical analysis?
8. Details description (methodology for the input parameters).
9. How the authors validated the model?
10. Input parameters for the goaf area?
11. Discuss the results comprehensively, taking relevant literature in citation.
12. The authors discussed the results in the conclusion. Your conclusion must be based on the results. Instead of discussing the results, show your conclusion.
13. No need the word research in the title.
Round 2
Reviewer 2 Report
1. The title has revised but not revised enough to show uniqueness. In present form, it is just a statement.
2. Map is not according to the l standard. Include the country map and then show the specific location in zoom.
3. Some contents are missing in Figure 6 (legend).
4. Units in Figure 10.
Author Response
1. The title has revised but not revised enough to show uniqueness. In present form, it is just a statement.
Authors’ Response:We agree with the reviewer’s opinion and have made the modification in the MS. Please see Lines 2 in the revised MS.
2. Map is not according to the l standard. Include the country map and then show the specific location in zoom.
Authors’ Response:We appreciate reviewer’s comments. We have tried to insert a map of China, but because the picture is too large, it needs to be compressed when inserted into MS, which will cause the picture is not clear. So we took a part of the map of China that was centered around Shanxi Province, please see he Figure 1(a) in the revised MS. And add a relevant description of the project location in the text section. Please see Lines 58-61 in the revised MS. I hope you are satisfied with this modification.
3. Some contents are missing in Figure 6 (legend).
Authors’ Response:We agree with the reviewer’s opinion and have made the modification in the MS. Please see the Figure 6 in the revised MS.
4. Units in Figure 10.
Authors’ Response:We agree with the reviewer's opinion and have made the modification in the MS. Please see the Figure 10 in the revised MS.